# Diffusion Reflection Measurements of Antibodies Conjugated to Gold Nanoparticles as a Method to Identify Cutaneous Squamous Cell Carcinoma Borders

**DOI:** 10.3390/ma13020447

**Published:** 2020-01-17

**Authors:** Asaf Olshinka, Dean Ad-El, Elena Didkovski, Shirel Weiss, Rinat Ankri, Nitza Goldenberg-Cohen, Dror Fixler

**Affiliations:** 1Department of Plastic Surgery, Rabin Medical Center—Beilinson Hospital, Petach Tikva 4941492, Israel; asafol@clalit.org.il (A.O.); Deana@clalit.org.il (D.A.-E.); 2Sackler Faculty of Medicine, Tel Aviv University, Tel Aviv 6997801, Israel; elenadi@clalit.org.il (E.D.); wshirel@gmail.com (S.W.); 3Department of Pathology and Cytology, Rabin Medical Center—Beilinson Hospital, Petach Tikva 4941492, Israel; 4The Krieger Eye Research Laboratory, Felsenstein Medical Research Center, Petach Tikva 49100, Israel; 5Faculty of Engineering and Institute of Nanotechnology and Advanced Materials, Bar Ilan University, Ramat Gan 5290002, Israel; rinnat8@gmail.com; 6Ruth and Bruce Rappaport Faculty of Medicine, The Technion—Technical Institute of Israel, Haifa 3200003, Israel; 7Department of Ophthalmology, Bnai Zion Medical Center, Haifa 3339419, Israel

**Keywords:** diffusion reflection, squamous cell carcinoma, epidermal growth factor receptor, gold nanoparticles, hyperspectral imaging

## Abstract

Diffusion reflectance spectroscopy measurements targeted with gold nanoparticles (GNPs) can identify residual cutaneous squamous cell carcinoma (SCC) in excision borders. Human SCC specimens were stained with hematoxylin and eosin to identify tumor borders, and reflected onto an unstained deparaffinized section. Diffusion reflection of three sites (normal and SCC) were measured before and after GNPs targeting. Hyperspectral imaging showed a mean of 2.5 sites with tumor per specimen and 1.2 tumor-free (*p* < 0.05, *t*-test). GNPs were detected in 25/30 tumor sites (sensitivity 83.3%, false-negative rate 16.6%) and 12/30 non-tumor sites (specificity 60%, false-positive rate 40%). This study verifies the use of nanotechnology in identifying SCC tumor margins. Diffusion reflection scanning has high sensitivity for detecting the residual tumor.

## 1. Introduction

The most frequent type of cancer in humans is skin cancer, and squamous cell carcinoma (SCC) is the second most frequent type of skin cancer [1]. The incidence of SCC has risen significantly in recent years [2]. SCC develops in the squamous epithelium-containing tissues of the skin and mucous membranes. Sun exposure is the most significant risk factor [3] because the ultraviolet B rays directly damage the epidermal DNA, which leads to genetic alterations that provide a selective growth advantage [4]. Other etiological factors in SCC development include genetics, exposure to radiation and chemicals, smoking, and human papillomavirus infection.

SCC is considered to be biologically aggressive, with a 12% rate of metastasis, especially for the lymph nodes [1]. The prognosis and natural history of the disease depend on tumor thickness, tumor location, and the degree of cellular differentiation. However, the main factor determining survival is completeness of the surgical resection [5,6]. In up to 22% of patients, the presence of residual gross or a microscopic tumor at the margins of resection is reported to substantially increase the local recurrence rate and decrease the survival rate [5,7,8,9]. Therefore, intraoperative evaluation for residual disease is essential [10,11]. At present, margins are routinely assessed during surgery by post-excision microscopic examination of the tissue (frozen section or Mohs’ procedure).

Like in many other cancers, we can find that the epidermal growth factor receptor (EGFR), a tyrosine kinase, or its ligands is overexpressed in SCC [2,12]. EGFR has an extracellular ligand-binding domain and a transmembrane region. Binding of the ligand to EGFR induces self-activation. In head and neck SCC, the common observation of EGFR overexpression suggests that the EGFR pathway is constitutively activated. This, in turn, activates signaling pathways responsible for cell proliferation, invasion, angiogenesis, survival, and metastasis [2], including mitogen-activated protein kinase (MAPK) cascade (RAS-RAF-MEK-MAPK), Phospholipase C-gamma/protein kinase C (PLC-gamma/PKC), phosphatidylinositol-3-OH kinase/AKT (PI-3K/AKT) and signal transducer and activator of transcription (STAT), or NFkappaB (v nuclear factor-κappa-Beta) which have all been found to be altered in SCC [13,14]. Activation of EGFR in keratinocytes, particularly with head and neck SCC, has been associated with epidermal thickness and cellularity, induction or inhibition of the keratinocyte differentiation (depending on the experimental conditions), the increase of cell survival and resistance to apoptosis, and the induction of keratinocytes migrating. It occurs as an early event in carcinogenesis in morphologically normal mucosa and increases with tumor progression. It seems to account for the acquisition of the tumor’s aggressive phenotype [4] and its often poor prognosis [15,16,17,18].

For the past several decades, metal nanoparticles (NPs) and especially noble metal nanomaterials are very attractive due to their superior properties and applications. They are used in a variety of applications such as a therapeutic agent delivery, photodynamic therapy, electronics, probes and sensors, catalysis, diagnostics, and more. Several methods have been established to synthesize metal NPs. Different types of templating methods are helpful in controlling the shapes and size such as metal-organic frameworks (MOFs), which are a series of porous structures that are usually constructed by organic linkers and metal joints in which metal NPs synthesized in different particle sizes. Evaporation induced self-assembly in emulsion using silica nanospheres as the hard template is another method. Synthesis of GNPs include chemical reduction, laser ablation, and vacuum sputtering. A chemical reduction, which is the most common way to produce GNPs, requires the use of reducing agents or stabilizing agents, which are sometimes toxic and not environmentally-friendly. We can find in the literature different methods, such as the microwave-induced plasma in a liquid process (PLP), which is described as scalable. A green technique to synthesize NPs, which does not need any reducing agents, utilized a ceramic coated electrode to prepare pure GNPs [19,20,21].

Increasing attention is being addressed to the use of gold nanoparticles (GNPs) for the selective diagnostic and imaging of many diseases and biological processes [22]. In particular, GNPs are a promising agent for the diagnosis and treatment of carcinoma [23,24].

According to the diffusion reflection model [25,26,27,28], light can be treated as a concentration of optical energy that diffuses down a concentration gradient. In irradiated tissue, the absorbing and scattering components causes the loss of energy within the tissue [25]. Together, the absorption and scattering coefficients constitute the tissue’s diffusion and reflection profiles [29]. GNPs have a unique shape and size-dependent optical properties. They have the ability to resonantly absorb and scatter visible and near-infrared (NIR) light on activation of their surface plasmons, which is shown spectroscopically as intense and narrow absorption/scattering peaks [30]. Studies have shown that GNPs can be distributed to a tissue of interest by conjugating them with an antibody specific to the tissue’s surface receptors. This causes a shift in the measured optical properties of the targeted tissue. By determining a difference in an absorption coefficient of a targeted (cancerous) tissue relative to normal tissue, the presence of the tumor can be detected [31].

GNPs are particularly suitable for this task due to their nontoxicity to living cells [32], high tissue penetration, and ease of generation, in addition to the safety of nonionizing radiation and reduced auto-fluorescence of the tissue in the NIR spectral range [33].

Prompted by findings that 90% of SCCs overexpress ERGF [18], our group recently investigated the use of reflectance spectroscopy for the detection of SCC targeted with GNPs conjugated to an anti-EGFR monoclonal antibody [25,26,28]. The aim of the present study is evaluation of the sensitivity of the method for identifying residual disease in borders of resected cutaneous SCC with a comparison to the histopathologic findings. Finding a novel method, a non-invasive matter using conjugated GNPs will be a good solution that can be used together with the practical surgical techniques and will give better clinical outcomes.

## 2. Materials and Methods

All methods were performed and approved in accordance with the relevant guidelines and regulations with the institutional and national review board.

The Ethical Committee of Rabin Medical Center approved the study protocol. The samples were taken unidentified from the pathology department, according to the pathological diagnosis, and, therefore, informed consent was not taken.

The study material consisted of 10 human SCC specimens that were excised in the department of plastic surgery of Rabin Medical Center in 2014. Two sequential 4-µm sections of each specimen were analyzed. The first specimen was stained with hematoxylin and eosin (H and E) for calibration of the methods (Figure 1). The H and E staining was reviewed by pathologists, who marked the tumor borders. The borders were then reflected from the H and E-stained section and marked on the second deparaffinized unstained section.

For the present study, we used gold nanospheres with a spectral peak of 530 nm and gold nanorods (GNRs) with a spectral peak of 517 and 670 nm. Different types of methods, based on previous research studies were used in controlling the size of the NPs. GNS were prepared following Enüstun and Turkevich [34] and the GNRs were prepared following Nikoobakht and El-Sayed [35].

Using transmission electron microscopy (TEM), the GNRs’ shape, size, and uniformity were characterized. Using a spectrophotometer, the GNRs extinction coefficient spectrum was determined, and the resultant extinction peak was 670 nm (Figure 2a). The resultant average shape was 25 × 11 ± 2.2 nm (Figure 2b).

Using a spectrophotometer, the GNSs extinction coefficient spectrum was determined, and the resultant extinction peak was 530 nm (Figure 2c). The resultant average shape was 20 ± 4.3 nm (Figure 2d). The GNRs bioconjugation to anti-EGFR was also validated by zeta potential and dynamic light scattering (DLS) measurements (Figure 2f).

Our selection of the described GNPs was for practical reasons. We used commercially available nanoparticles, which will help us with the implementation in the medical clinical field. Any different form that we will have to synthesize by ourselves will complicate the process with the regulator for approval for use with human patients.

The GNPs were conjugated to two types of anti-EGFR antibodies: recombinant human/mouse chimeric EGFR receptor monoclonal antibody (Cetuximab, Erbitux, Merck KGaA, Germany) and a platelet-derived growth factor receptor-α affinity purified rabbit polyclonal antibody (Santa Cruz Biotechnology, Santa Cruz, CA, USA). Bioconjugation of the GNPs to the anti-EGFR antibody was achieved using polystyrene sulfonate [36], according to the method described by Lvov. Each slide was scanned before and after adding the GNRs for a negative control (described in Figure 2e).

Tissue reflectance was measured with the Nuance hyperspectral imaging system (Nuance, CRi, Woburn, MA, USA) using a halogen light source (UN2-PSE100, Nikon, Japan) with a 40× objective (0.75 NA) and a 32-bit ultrasensitive charge-coupled device camera detector (N-MSI-EX) for imaging in the color codes chart red green blue (RGB) mode. Microscopy was performed with a Nikon 80i microscope. Images were acquired through Nuance 2.1 software (Burlington, MA, USA). Reflectance measurements are presented in arbitrary intensity units (IU). The following method was used. The reflectance spectra values were measured in the three sites identified microscopically as tumor tissue and the three sites identified microscopically as tumor-free tissue on each unstained slide. Each slide was scanned before and after GNPs targeting, at the same positions, and the background noise was subtracted. The results without GNPs were subtracted from the results with GNPs. The glass spectra were subtracted from the result to reduce spectral noise.

To visualize the GNPs in the tested tissue, we used the the airSEM^TM^, which is an innovative high-resolution scanning electron microscope (B-nano Ltd., Rehovot, Israel) to scan the GNPs-stained slides. The airSEM operates in open air, which bypasses obstacles associated with vacuum-based systems, such as electron scatter by gas molecules. This makes signal collection more efficient, at a resolution better than 5 nm under ambient conditions. Operation principles of airSEM microscope (Figure 3) include the following. As in any other electron microscope (EM), electrons are emitted from an electron source in vacuum (a field emitter scanning electron microscope (SEM) column) and are then collimated and focused using a standard electron optics apparatus to form an ultra-sharp beam. In contrast to standard electron microscope (EM) techniques, where the sample resides in the same vacuum environment shared with a lower part of the column in airSEM, the sample resides in air under ambient conditions.

For statistical analysis, we used paired two-sample Student’s *t*-test for means. *p* < 0.5 was considered statistically significant.

## 3. Results

Findings on H and E staining with light microscopy and on hyperspectral imaging were compared, and we found that most of the histologically proven cancer sites were detected on hyperspectral imaging. The higher measured intensity in the tumor tissue relative to the normal tissue made it easy to discriminate between cancerous and non-cancerous tissue.

Figure 1 shows the histological morphology of an SCC in a slide stained by H and E. Figure 4 shows a representative SCC specimen after staining with a GNPs-conjugated EGFR antibody (The location of the GNRs is marked in red.). We found a correlation between the SCC contained tissue and the reflectance spectra intensity readings from the GNPs stained slides. A representative reflectance spectra intensity profiles are presented in Figure 5 in which high reflectance was found at peak GNPs wavelength (670 nm) in tissue areas identified histologically as SCC. International units (IU) values were lower in areas of the non-tumor tissue.

For each of the 10 specimens, we evaluated the sites found to contain the tumor by hyperspectral imaging out of the sites shown microscopically to contain the tumor. The sites shown microscopically were found to be tumor-free. Corresponding mean values were 2.5/3 for tumor tissue and 1.2/3 for non-tumor tissue. The difference between the normal area and tumor area was statistically significant (*p* < 0.05).

GNPs infiltrated the entire tumor samples. We detected GNPs in 25 of the total 30 tumor-containing tissue sites, for a sensitivity of 83.3% and a false-negative rate of 16.6%, and in 12 of the 30 tumor-free tissue sites, for a specificity of 60% and a false-positive rate of 40%. Figure 6 shows the visibly higher concentration of GNP particles in the tumor-containing tissue than the tumor-free tissue by high-resolution electron microscopy. To further confirm the presence of GNPs in the tissue containing SCC, we formulated a photon-detector graph in which a clear peak was observed at the gold (Au) region of the spectrum. All other detected elements were mostly components of the carrier glass. We obtained the same results with the gold nanospheres (GNSs) (data not shown) but we specifically used GNR (and not GNS) since GNRs exhibit unique absorption properties in the near-infrared (NIR) region, where light penetration through the tissues is relatively high (up to a few cm).

## 4. Discussion

We describe the application of a novel nanotechnology using a biological marker to detect clusters of cells left behind after tumor-removal surgery. Hyperspectral imaging of post-surgical specimens of SCC stained with GNPs conjugated to EGFR antibodies revealed most of the histologically-proven cancer sites. By measuring the intensity scattered from the GNPs, we could easily distinguish the cancerous tissue from the neighboring normal tissue. This is the first successful use of this technique for the post-operative evaluation of human SCC specimens. It follows earlier experiments in mice and human oral SCC [26,28]. At present, clinically, the most effective treatment for a remaining cancerous tissue after surgery or positive surgical margins is re-operation with excision of additional tissue [5], or adjuvant radiotherapy. If our technique is found feasible for intraoperative use, it could dramatically improve the odds against recurrence and spare patients improve further invasive interventions.

The standard surgical approach to SCC is surgical excision of the primary cancerous lesion. The surgical margins’ integrity is evaluated by multiple intraoperative samplings of the resultant defect periphery for frozen section analysis [7] or analysis of the tissue excision margins as part of Mohs’ surgery (in which tumor margins sampling is potentially more thorough) [8]. However, in 70% of head and neck cancers, microscopic “fingers” of the tumor that are 10 to 20 cells wide extend at least 1 cm away from the gross disease. These minute extensions may be missed with standard sampling techniques [8], which substantially reduce surgical control of the disease and, therefore, patient survival [7,9].

Most of the few currently available methods for identifying residual cancer at the resection bed are fluorescence-based. They are not directed at the cancer itself [37,38,39,40] and provide only indirect estimated differences of tissue illumination and reflection between cancerous and surrounding normal tissue. Their main disadvantages are background fluorescence noise and low contrast. Furthermore, since in vivo visualization of tissue fluorescence in the bed may be problematic, these techniques have been limited mainly to resected and archival tissue.

Gleysteen et al. [41] fluorescently labeled anti-EGFR antibody with an NIR fluorophore (Cy5.5). They used mice following intravenous or intraoral injections of head and neck SCC and were able to detect the cells. They concluded that cetuximab-Cy5.5 may have clinical utility in detecting and guiding the removal of regional and distant micro-metastasis.

Heath et al. [42] recently reported that panitumumab, which is a monoclonal antibody-targeting EGFR, can be used in real-time fluorescent imaging with 100% sensitivity on the histologic processing of cutaneous SCC in mice. An NIR scanning device (Odyssey) was used to measure fluorescence intensity in histological sections. Tumor detection was substantially improved with panitumumab-IRDye800 compared with IgG-IRDye800.

Our study was designed to evaluate the sensitivity of direct reflectance scanning for the detection of tumor cells in tissue and excision borders in histological sections using anti-EGFR-conjugated GNPs. The superior absorption properties of GNPs have already been utilized for photothermal therapy [43] and other methods of selective cancer therapy, as demonstrated in glioblastomas [40,41], even though there is risk of damage to neighboring tissue. Previous studies also presented diffuse reflectance measurements for a cancer diagnosis [44,45,46], but they did not use nanoparticles as contrast agents. Recently, a novel strategy to treat cancer cells expressing a higher level of the EGFR used an epidermal growth factor-gold nanoparticle (EGF-GNP) conjugate complex [47]. Examination of cell levels showed that the EGF-GNP conjugated complex elicited phosphorylation of extracellular signal-regulated kinases (ERK) [47], and elevated EGFR expression, which led to internalization of the GNP into the cells. Treatment by non-thermal plasma irradiation increased apoptosis. Kim et al. [47] concluded that non-thermal plasma irradiation coupled with EGF-GNP treatment can trigger DNA damage, which led to cell death. The present study confirms the value of GNPs as contrast agents because the reflection measurement is directed at their unique absorption properties in the NIR light region rather than their scattering properties. Therefore, no contrast interruptions are expected. This makes our technique suitable for in vivo use. EGFR served as the biological marker because it is highly overexpressed in SCC as well as in other human cancers. Accordingly, EGFR is frequently used as a therapeutic target [18,48,49,50]. In addition, studies have shown that increased EGFR protein expression and EGFR gene copy number amplification are associated with a poor prognosis [18,51]. The results verify the sensitivity and efficiency of this technique. The size-dependent and shape-dependent optical properties of the GNPs covalently attached to EGFR antibodies changed the optical properties of the cancerous cells into which they were absorbed, which makes the cells easily identifiable on hyperspectral imaging, even in apparently tumor-free tissue.

The technique described is nonionizing, noninvasive, and easy to perform. Given that measurements are objective and not dependent on a qualified technician, fewer interpretation errors are expected. Our group [52,53] evaluated the detection sensitivity of reflection measurements of GNRs bio-conjugated to EGFR (GNRs-EGFR) monoclonal antibodies. Tissue sections incubated with GNRs-EGFR and the reflectance spectrum was measured using hyperspectral microscopy. The results showed that reflectance intensity increased with the progression of the disease. The GNRs reflection measurements can discriminate benign and mild dysplastic lesions from the more severe dysplastic and invasive cancer lesion, which suggests an objective technique, not dependent on the qualification of a technician with fewer interpretation errors.

## 5. Conclusions

This study shows that the hyperspectral signal may be used to detect a residual tumor in post-surgical specimens of cutaneous SCC stained with anti-EGFR conjugated GNPs. The findings were confirmed by comparing to standard pathological staining. The technique offers great promise as an intraoperative tool for identifying patients with positive margins who may benefit from the need of reoperation or radiotherapy. Moreover, the follow-up in patients found to have clean surgical margins by this analysis might be limited to tight clinical follow-up only. Further studies using in vivo models are required to validate the results of this study.

## 6. Summary Points


Squamous cell carcinoma (SCC) is the second most frequent type of skin cancer.SCC is considered to be biologically aggressive, with a 12% rate of metastasis, especially to the lymph nodes.The main factor determining survival is completeness of the surgical resection.SCC is characterized by overexpression of epidermal growth factor receptor (EGFR).GNPs are a promising agent for the diagnosis and treatment of carcinoma, and, in this case, were conjugated to EGFR.This study investigated the ability to detect residual SCC stained with gold nanoparticles using diffusion reflectance.This study verifies the use of nanotechnology in identifying SCC tumor margins.The noninvasive, nonionizing direct diffusion reflection scanning has high sensitivity for distinguishing cancerous tissue histologically.Diffusion reflection is a promising tool for the intraoperative identification of residual disease in SCC excision borders.


## Figures and Tables

**Figure 1 materials-13-00447-f001:**
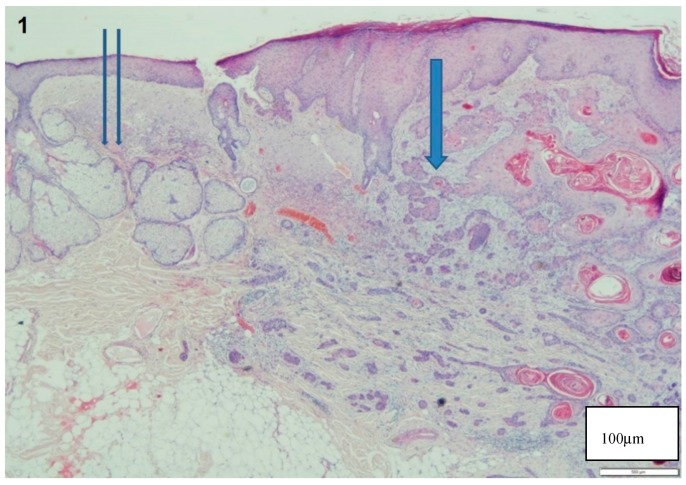
Histological findings of SCC in H and E-stained slide. Note the invasion of dermis by atypical keratinocytes with squamous eddies and pearls and dyskeratosis in the tumor (**arrow**) and the normal tissue (**2 arrows**).

**Figure 2 materials-13-00447-f002:**
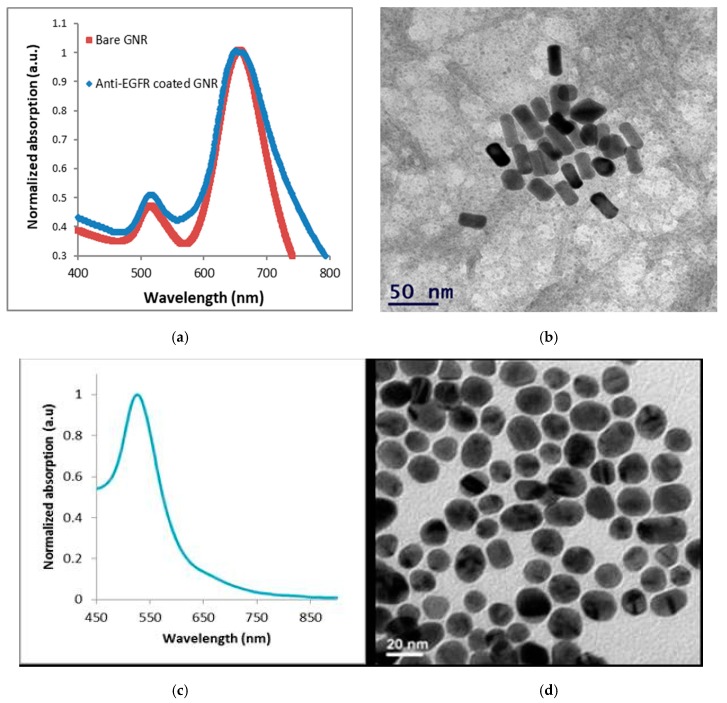
Characterization of GNRs. (**a**) Optical properties of the GNRs: ultraviolet-visible spectroscopy of bare GNRs and anti-EGFR coated GNRs. The resulting absorption peak was 670 nm. (**b**) Transmission electron microscopy image of the GNRs. Average dimensions were 25 × 11 ± 2.2 nm (n = 10). (**c**) Optical properties of the GNSs: ultraviolet-visible spectroscopy of anti-EGFR coated GNSs. The resulting absorption peak was 530 nm. (**d**) Transmission electron microscopy image of the GNRs. Average dimensions were 20 ± 4.3 nm (n = 8). (**e**) Schematic diagram of the GNRs synthesis process and their coating with m-PEG (85%) and COOH-PEG (15%), which is followed by a covalent conjugation to anti-EGFR. (**f**) Zeta potential and dynamic light scattering (DLS) size measurements (at 250 °C) of bare GNRs and anti-EGFR coated GNRs coated. The significant difference that was obtained (by zeta potential, DLS, and UV-vis spectroscopy) following coating, which demonstrates the efficiency of the chemical coating.

**Figure 3 materials-13-00447-f003:**
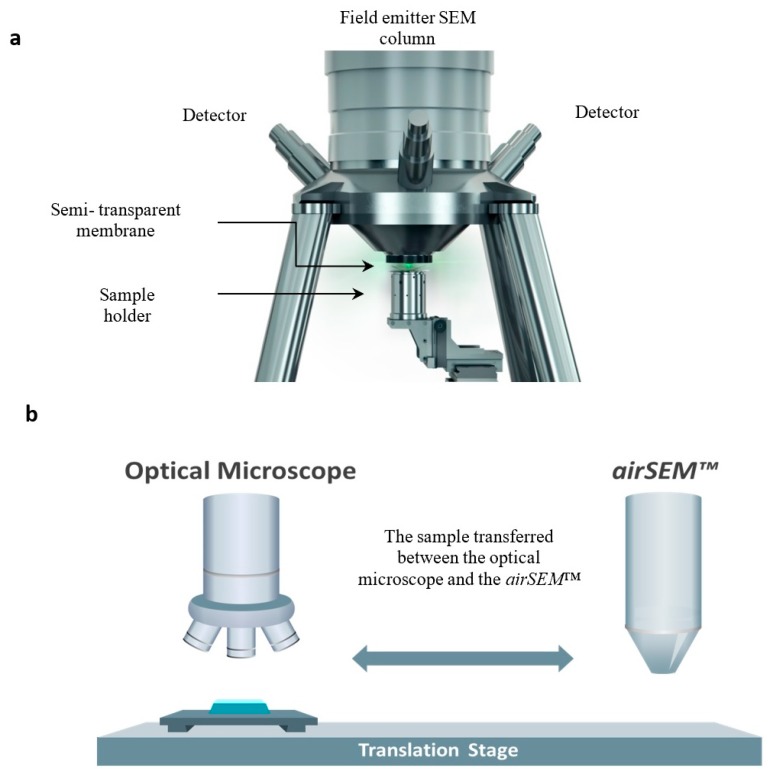
Schematic view of the air scanning electron microscope (airSEM) (**a**) and an imaging station operating principle (**b**). The two visualizing modalities are combined onto one platform. The sample is shuttled between the optical microscope and airSEM, which provides accurate navigation and continuation of magnification of the region of interest (ROI).

**Figure 4 materials-13-00447-f004:**
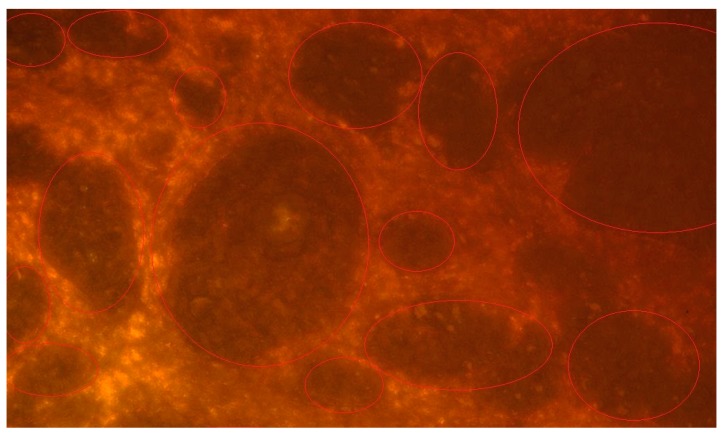
Hyperspectral image of the SCC specimen after staining with EGFR-conjugated GNPs. The location of the GNRs are marked in red.

**Figure 5 materials-13-00447-f005:**
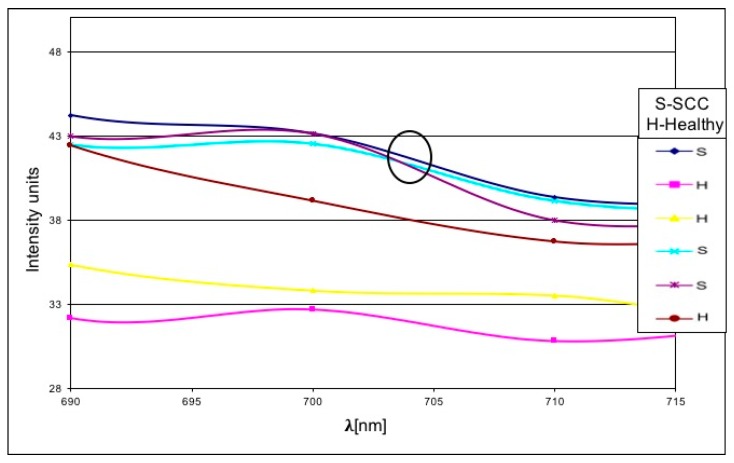
Reflectance spectra values in three tumor sites and three tumor-free sites. Focused view of second peak wavelength of the particle measured in the graph. Note the values in the area of the SCC.

**Figure 6 materials-13-00447-f006:**
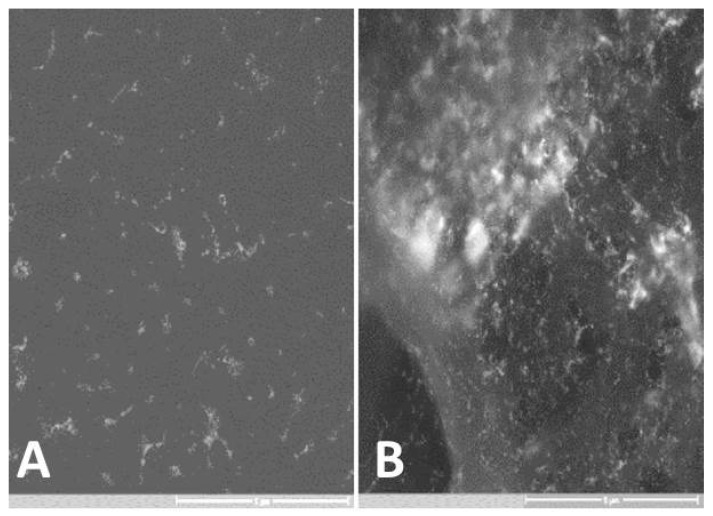
GNP-stained slide viewed with a high resolution (X19K) scanning electron microscope (airSEM™). A significantly lower density of GNPs can be seen in the tumor-free tissue (**A**) in comparison to the SCC-containing tissue (**B**).

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
