# Peer review of "Diffusion Reflection Measurements of Antibodies Conjugated to Gold Nanoparticles as a Method to Identify Cutaneous Squamous Cell Carcinoma Borders"

_materials, 2020, doi:10.3390/ma13020447_

Round 1

Reviewer 1 Report

The authors describe the use of gold nanorods to identify the borders of squamous cell carcinoma. The article is well written, the experimental details are well described, the scientific research is sound and the conclusions are adequate. Overall, I think it is a very good article that can be of interest to many researchers.

There are a few minor corrections that need to be done before publication. 1) 3rd and 4th paragraphs seem to be instructions for authors unrelated to the article, and thus should be removed. 2) Line 107: the authors state that they have used gold nanospheres and gold nanorods, but show no results or characterization of gold nanospheres.

Author Response

Dear Reviewer 1,

Thank you for giving us the opportunity to submit a revised draft of our manuscript titled Diffusion Reflection Measurements of Antibodies Conjugated to Gold Nanoparticles as a Method to Identify Cutaneous Squamous Cell Carcinoma Borders to Materials. We appreciate the time and effort that you and the reviewers have dedicated to providing your valuable feedback on our manuscript. We are grateful to the reviewers for their insightful comments on this paper. We have been able to incorporate changes to reflect most of the suggestions provided by you and the reviewers. We have marked the changes within the manuscript.

Here is a point-by-point response to your comments and concerns.

Comment 1: 3rd and 4th paragraphs seem to be instructions for authors unrelated to the article, and thus should be removed.

Response: Agree. We have, accordingly, removed this part. Thank you.

Comment 2: Line 107: the authors state that they have used gold nanospheres and gold nanorods, but show no results or characterization of gold nanospheres.

Response: Thank you for pointing this out. We incorporated a clarification text in the results section (line 233)

Thank you again for your comments; it truly helped us to improve our manuscript.

Reviewer 2 Report

This manuscript reports the use of EGFR conjugated gold nanorods for diffusion reflectance spectroscopy to identify SCC in human specimens. However, it is not clear what are the advantages and novelty of using gold nanorods for the detection and the false positive rate of 40% and false negative rate of 16% both seem very high, hinting the non-specificity of this method. Some specific comments are as follows:

Page 3, line 91-95, this paragraph seems to be duplicated from template. Please revise or remove that. Page 4, line 107, I don’t see any result using gold nanoparticles, and there is no comparison made to show whether nanoparticle or nanorod is better so there is no need to show this. The validation of bioconjugation should be presented with at least the inclusion of zeta potential results. Figure 1 & 4 should also show the SCC sample vs. the normal tissue sample side-by-side for comparison. Especially, you should annotate Figure 4 to show where the gold rods are. For Figure 5, I think the spectra of all SCC and normal samples should be shown and a careful analysis should be clearly elaborated to let the readers know how the spectral data are interpreted. The tumor-free sites shown are vastly different in terms of their spectra, therefore I am not very convinced that the gold nanorod staining is even specific.

Author Response

Dear Reviewer 2,

Thank you for giving us the opportunity to submit a revised draft of our manuscript titled Diffusion Reflection Measurements of Antibodies Conjugated to Gold Nanoparticles as a Method to Identify Cutaneous Squamous Cell Carcinoma Borders to Materials. We appreciate the time and effort that you and the reviewers have dedicated to providing your valuable feedback on our manuscript. We are grateful to the reviewers for their insightful comments on this paper. We have been able to incorporate changes to reflect most of the suggestions provided by you and the reviewers. We have marked the changes within the manuscript.

Here is a point-by-point response to your comments and concerns.

Comment 1: The advantages and novelty of using gold nanorods for the detection and the false positive rate of 40% and false negative rate of 16% both seem very high, hinting the non-specificity of this method.

Response: You have raised an important point here, thank you. We believe that gold nanorods are particularly suitable for this task owing to their nontoxicity to living cells, ease of generation, and relatively high tissue penetration, in addition to the safety of nonionizing radiation and reduced auto-fluorescence of the tissue in the NIR spectral range. Skin cancer is a very common pathology, in which the margin detection methods (pre-surgical and surgical) are very inaccurate. Finding a novel method, non-invasive, with false-positive rate of 40% is a good solution that can be used together with clinical assessment and surgical technique and will give better numbers.

Comment 2: Page 3, line 91-95, this paragraph seems to be duplicated from template. Please revise or remove that.

Response: Agree. We have, accordingly, removed this part. Thank you.

Comment 3:  Page 4, line 107, I don’t see any result using gold nanoparticles, and there is no comparison made to show whether nanoparticle or nanorod is better so there is no need to show this

Response: We thank the reviewer for his comment and apologize for the lack of clarity. We incorporated a clarification text in the results section (line 233).

Comment 4:  The validation of bioconjugation should be presented with at least the inclusion of zeta potential results.

Response: We agree with this and have incorporated your suggestion throughout the manuscript at the m&m section (lines 166-169)

Comment 5:  Figure 1 & 4 should also show the SCC sample vs. the normal tissue sample side-by-side for comparison. Especially, you should annotate Figure 4 to show where the gold rods are

Response: Thank you for this suggestion. However, in the case of our study, it seems slightly out of scope because comparing figure 1 to a normal histological slide will not show the borders of the tumor as this will not be continuous tissue. Regarding figure 4, we agree, and accordingly, marked it.

Comment 6:  For Figure 5, I think the spectra of all SCC and normal samples should be shown and a careful analysis should be clearly elaborated to let the readers know how the spectral data are interpreted.

Response: We appreciate your comments. The rationale of presenting figure 5 as followed is for demonstrating representative reflectance spectra intensity profiles that were measured in our experiment; on each unstained slide, the reflectance spectra values were measured in 3 sites of the part identified as tumor tissue and the 3 sites at the part identified as tumor-free tissue. Each slide was scanned before and after GNPs targeting, at the same positions, and the background noise was subtracted. The results without GNPs were subtracted from the results with GNPs. The glass spectra were subtracted from the result to reduce spectral noise. Each slide was scanned before and after adding the GNRs for negative control. We found a correlation between the SCC contained tissue and the reflectance spectra intensity readings

Thank you again for your comments; it truly helped us to improve our manuscript.

Reviewer 3 Report

Overall:

This manuscript reports diffusion reflectance spectroscopy by using Au nanoparticles (AuNPs) that can identify SCC efficiently. This paper is interesting and promising, and the physical chemistry properties were well characterized. Although there have been many reports dealing with the similar metal nanomaterials, the report here is useful for readers working on the related fields. I recommend accepting it for publication after minor revision. My comments are as follows.

The introduction of metal nanomaterials can be further improved. Several recent papers are given below and the authors are suggested to cite these papers. (a) The Ariga group: Au nanoparticles: Journal of Nanoscience and Nanotechnology. 2016, 16, 9257-9262. (b) The Wu group: ChemCatChem. 2016, 8, 506-509. The role of Au nanoparticles should be further discussed. For examples, what is the advantage of nanoparticles, instead of nanowires or other morphologies. Please also give more details on how to control the particle size of Au NPs. Figure 3 seems to be just a photo. It would be better if the authors can provide the corresponding illustration that also show the function of each component in the machine. It is still difficult to find out the Au NPs from the Figure 4. Is there any other way to do the staining or without staining. Figure 6 can be improved. The quality is not good.

Author Response

Dear Reviewer 3,

First, we would like to thank the reviewer for his positive opinion and for supporting the publication of this paper.

Thank you for giving us the opportunity to submit a revised draft of our manuscript titled Diffusion Reflection Measurements of Antibodies Conjugated to Gold Nanoparticles as a Method to Identify Cutaneous Squamous Cell Carcinoma Borders to Materials. We appreciate the time and effort that you and the reviewers have dedicated to providing your valuable feedback on our manuscript. We are grateful to the reviewers for their insightful comments on this paper. We have been able to incorporate changes to reflect most of the suggestions provided by you and the reviewers. We have marked the changes within the manuscript.

Here is a point-by-point response to your comments and concerns.

Comment 1: The introduction of metal nanomaterials can be further improved. Several recent papers are given below and the authors are suggested to cite these papers. (a) The Ariga group: Au nanoparticles: Journal of Nanoscience and Nanotechnology. 2016, 16, 9257-9262. (b) The Wu group: ChemCatChem. 2016, 8, 506-509. (c) The Yonezawa group: Pt nanoparticles. Materials Letters.2016, 164, 488-492.

Response: We thank the reviewer for his question and apologize for the lack of clarity. We agree with this comment and think that it is a good idea to do so, therefore, we have discussed these publications in the manuscript (Line 63) and added them as new references (49-53).

Comment 2: The role of Au nanoparticles should be further discussed. For examples, what is the advantage of nanoparticles, instead of nanowires or other morphologies.

Response: We appreciate your comments. Our selection in the described gold nanoparticles was simply practical. We used "on the shelf" commercial available nanoparticles which will help us with the implementation in the medical clinical field. Any different form that we will have to synthesize by ourselves will complicate the process with the regulator for approval the use with human patients. We think it's an important point that you raised so we add this discussion in the manuscript (line 139).

Comment 3:  Please also give more details on how to control the particle size of Au

Response: We thank the reviewer for his comment, therefore, we add a discussion in the text (line 110) and references: GNS were prepared following Enüstun and Turkevich. (Enustun BV, Turkevich J. Coagulation of Colloidal Gold. Journal of the American Chemical Society. 1963/11/01 1963;85(21):3317-3328.)  The GNRs were prepared following Nikoobakht and El-Sayed. (B. Nikoobakht, M. A. El-Sayed. Preparation and Growth Mechanism of Gold Nanorods (NRs) Using Seed-Mediated Growth Method. Chem. Mater. 2003;15:1957-1962.)

Comment 4: Figure 3 seems to be just a photo. It would be better if the authors can provide the corresponding illustration that also shows the function of each component in the machine.

Response: Agree. We have, accordingly, changed the illustration and the legend of the figure. Thank you for pointing this out, it helped make it more understandable.

Comment 5: It is still difficult to find out the Au NPs from the Figure 4. Is there any other way to do the staining or without staining.

Response: We thank the reviewer for his important comments. On figure 6 the NPs can be seen in the tumor-free tissue and we addressed it in the manuscript and marked the stained EGFR-conjugated NPs in figure 4.

Comment 6: Figure 6 can be improved. The quality is not good.

Response: Thank you for pointing this out. We will improve the figure resolution.

Thank you again for your comments; it truly helped us to improve our manuscript.

Round 2

Reviewer 2 Report

Thank you for taking into consideration most of my comments. A few more points to address:

I think it would be good to clarify your response to my comment 1 in the introduction of the manuscript. I believe other readers will have the same queries in mind. Figure 2 a-d, can you rescale the subfigures to make them of equal sizes? In the response, the authors mentioned that you have marked Figure 4, but I don't see any markings. Can you clarify?

Author Response

Reviewer 2 evaluation

Thank you for giving us the opportunity to submit a revised draft of our manuscript titled Diffusion Reflection Measurements of Antibodies Conjugated to Gold Nanoparticles as a Method to Identify Cutaneous Squamous Cell Carcinoma Borders to Materials. We appreciate the time and effort that you dedicated to providing your valuable feedback on our manuscript in extremely short period of time (less than 24 hours!). We have been able to incorporate changes to reflect most of the suggestions provided by you. We have marked the changes within the manuscript.

Here is a point-by-point response to your comments and concerns.

Comment 1: I think it would be good to clarify your response to my comment 1 in the introduction of the manuscript. I believe other readers will have the same queries in mind.

Response: You have raised an important point here, thank you.  We have, accordingly, incorporated a clarification text in the introduction section (line 98).

Comment 2: Figure 2 a-d, can you rescale the subfigures to make them of equal sizes?

Response: Agree. We have, accordingly, changed the figures. Thank you.

Comment 3: In the response, the authors mentioned that you have marked Figure 4, but I don't see any markings. Can you clarify?

Response: Thank you for pointing this out. We have, accordingly, marked the figure. Thank you.

Thank you again for your comments; it truly helped us to improve our manuscript.
